# Lithium: A Promising Anticancer Agent

**DOI:** 10.3390/life13020537

**Published:** 2023-02-15

**Authors:** Edgar Yebrán Villegas-Vázquez, Laura Itzel Quintas-Granados, Hernán Cortés, Manuel González-Del Carmen, Gerardo Leyva-Gómez, Miguel Rodríguez-Morales, Lilia Patricia Bustamante-Montes, Daniela Silva-Adaya, Carlos Pérez-Plasencia, Nadia Jacobo-Herrera, Octavio Daniel Reyes-Hernández, Gabriela Figueroa-González

**Affiliations:** 1Unidad Multidisciplinaria de Investigación Experimental Zaragoza, Facultad de Estudios Superiores Zaragoza, Universidad Nacional Autónoma de México, Ciudad de México 09230, Mexico; 2Unidad de Estudios Superiores Tultitlán, Universidad Mexiquense del Bicentenario, Ocoyoacac 54910, Mexico; 3Laboratorio de Medicina Genómica, Departamento de Genómica, Instituto Nacional de Rehabilitación Luis Guillermo Ibarra Ibarra, Ciudad de México 14389, Mexico; 4Facultad de Medicina, Universidad Veracruzana, Ciudad Mendoza 94740, Mexico; 5Departamento de Farmacia, Facultad de Química, Universidad Nacional Autónoma de México, Ciudad de México 04510, Mexico; 6Licenciatura en Médico Cirujano, Facultad de Ciencias de la Salud Universidad Anáhuac Norte, Academia de Genética Médica, Naucalpan de Juárez 52786, Mexico; 7Escuela Superior de Medicina, Instituto Politécnico Nacional, Ciudad de México 11340, Mexico; 8Facultad de Medicina, Universidad Autónoma del Estado de México, Toluca de Lerdo 50180, Mexico; 9Laboratorio Experimental de Enfermedades Neurodegenerativas, Instituto Nacional de Neurología y Neurocirugía, Ciudad de México 14269, Mexico; 10Laboratorio de Genómica, Instituto Nacional de Cancerología (INCan), Ciudad de México 14080, Mexico; 11Laboratorio de Genómica, Unidad de Biomedicina, Facultad de Estudios Superiores Iztacala, Universidad Nacional Autónoma de México, Tlalnepantla 54090, Mexico; 12Unidad de Bioquímica, Instituto Nacional de Ciencias Medicas y Nutrición Salvador Zubirán (INCMNSZ), Ciudad de México 14080, Mexico; 13Laboratorio de Biología Molecular del Cáncer, Unidad Multidisciplinaria de Investigación Experimental Zaragoza, Facultad de Estudios Superiores Zaragoza, Universidad Nacional Autónoma de México, Ciudad de México 09230, Mexico; 14Laboratorio de Farmacogenética, Unidad Multidisciplinaria de Investigación Experimental Zaragoza, Facultad de Estudios Superiores Zaragoza, Universidad Nacional Autónoma de México, Ciudad de México 09230, Mexico

**Keywords:** lithium, cancer, apoptosis, autophagy, anti-cancer effects, nano-delivery

## Abstract

Lithium is a therapeutic cation used to treat bipolar disorders but also has some important features as an anti-cancer agent. In this review, we provide a general overview of lithium, from its transport into cells, to its innovative administration forms, and based on genomic, transcriptomic, and proteomic data. Lithium formulations such as lithium acetoacetate (LiAcAc), lithium chloride (LiCl), lithium citrate (Li_3_C_6_H_5_O_7_), and lithium carbonate (Li_2_CO_3_) induce apoptosis, autophagy, and inhibition of tumor growth and also participate in the regulation of tumor proliferation, tumor invasion, and metastasis and cell cycle arrest. Moreover, lithium is synergistic with standard cancer therapies, enhancing their anti-tumor effects. In addition, lithium has a neuroprotective role in cancer patients, by improving their quality of life. Interestingly, nano-sized lithium enhances its anti-tumor activities and protects vital organs from the damage caused by lipid peroxidation during tumor development. However, these potential therapeutic activities of lithium depend on various factors, such as the nature and aggressiveness of the tumor, the type of lithium salt, and its form of administration and dosage. Since lithium has been used to treat bipolar disorder, the current study provides an overview of its role in medicine and how this has changed. This review also highlights the importance of this repurposed drug, which appears to have therapeutic cancer potential, and underlines its molecular mechanisms.

## 1. Introduction

Cancer is a disease characterized by the uncontrolled growth of certain cells. The phenotypic and genotypic complexity of cancer cells translates into the acquisition of functional capabilities, to continue to grow and spread. Capabilities for sustaining proliferative signaling, evading growth suppressors, non-mutational epigenetic reprogramming, avoiding immune destruction, enabling replicative immortality, tumor-promoting inflammation, polymorphic microbiomes, activating invasion and metastasis, inducing or accessing vasculature, senescent cells, genome instability and mutation, resisting cell death, deregulating cellular metabolism, and unlocking phenotypic plasticity are all hallmarks of cancer [1]. Metabolic alterations in cancer, known as the Warburg effect [2], have emerged as an alternative therapeutic pathway to deprive the body and tumor of glucose [3]. Therefore, there has been interest in ketogenic bodies (KBs), to evaluate the mechanism of glucose deprivation as a therapy in cancer. Among the KBs, acetoacetate (AcAc) is the most abundant and relevant [3] and, for in vitro and in vivo assays, AcAc is often used in the form of lithium acetoacetate (LiAcAc). Lithium has various therapeutic effects on neurotrophic factors, neurotransmitters, oxidative metabolism, apoptosis, neuronal structures, and glia [4,5]. In this context, several publications reported that LiAcAc affects cancer cell growth. Other reports, however, indicated that lithium chloride (LiCl) inhibited cell growth in a dose-dependent manner, similarly to LiAcAc. Several questions arise in this regard, is lithium involved in certain hallmarks of cancer? Does lithium have a therapeutic potential in cancer? In this review, genomic, transcriptomic, and proteomic approaches are used to try to answer these questions.

## 2. The Effect of Lithium on Cancer Progression: Preclinical Approach

Lithium is a highly reactive alkali metal that exists as Li^+^ in seawater and in minerals [6]. The Li^+^ cation is therapeutically relevant, given that it has been used to treat bipolar disorders [7,8,9], Alzheimer’s disease [10,11], and as adjunctive therapy in thyroid cancer [12,13]. For bipolar mood disorders, achievable Li levels in plasma are around 0.5–1.0 mM (mEq/L) [14], whereas concentrations higher than 2 mM cause cytotoxicity in patients, resulting in nausea, fine tremors, diarrhea, confusion, slurred speech, and ataxia [14]. Lithium forms a complex with ATP and Mg^2+^, resulting in the bioactive form of lithium that modulates purine receptor activity in neuronal cells [15]. Lithium had been used over decades for bipolar disorder treatment, preventing relapses, and for the prophylaxis of manic episodes [16]. Some morphological abnormalities have been observed in patients with bipolar disorder. The neuroprotective activity of Li is related to the inhibition of its molecular targets, such as the enzymes inositol-monophosphate (IMPase), glycogen-synthase-kinase 3β (GSK3), and protein kinase C (PKC) [17]. Interestingly, these enzymes are deregulated in cancer, suggesting a potential role for Li, beyond bipolar disorders, in cancer treatment. Recent in vitro and in vivo studies reported the effects of lithium (lithium acetoacetate (LiAcAc), lithium chloride (LiCl), lithium citrate (Li_3_C_6_H_5_O_7_), and lithium carbonate (Li_2_CO_3_) on cell proliferation, tumor growth, and cancer metabolism. According to this evidence, LiAcAc inhibited cell proliferation in colon, breast, pancreatic, and neuroblastoma cancers [18]. Table 1 shows some recent preclinical studies elucidating the value of lithium in cancer treatment.

## 3. Lithium Transport

The Li^+^ ion shares transport and permeation pathways with other ions, such as sodium [38]. Therefore, extracellular and intracellular concentrations of the Li^+^ ion depend on its ingestion [39]. However, LiAcAc requires specific transporters. In cells, the presence of monocarboxylate transporters (MCTs) is involved in the passive transport of lactate, pyruvate, D-β-hydroxybutyrate, and acetoacetate across the cell membrane [40]. MCT2 has the highest affinity constant (Km) for KBs such as D-β-hydroxybutyrate and acetoacetate [41], but these substrates also are transported by MCT1 [42] and MCT4, which have a lower affinity for them [43]. LiAcAc is imported into cells through MCT1 and MCT2, with Km values of 5.5 mM and 0.8 mM, respectively [41,43,44]. These values indicate that MCT2 has a higher affinity for LiAcAc compared to MCT1, therefore the cellular expression of MCT2 could determine the cellular response to Li^+^ ion exposure. Differential expression of MCTs in healthy and cancer tissues has been reported. In this way, MCT1 is expressed by healthy or cancer tissues [45]. Furthermore, MCT1 has a specific expression in cardiomyocytes, hepatocytes, and cells of the gastrointestinal tract (stomach, duodenum, jejunum, ileum, cecum, colon, and rectum) [44,46,47,48,49,50,51,52,53,54,55]. In addition, this transporter is found in erythrocytes, lymphocytes, and monocytes [46,56], in adipocytes [57,58] and in mammary glands [59]. Moreover, MCT1 is found in endothelial brain cells, astrocytes, oligodendrocytes, microglial cells, and the hypothalamus [60,61,62,63,64]. Furthermore, this transporter was found in the peripheral nervous system [65,66]. While MCT1 is a highly distributed transporter, MCT2 has a restricted distribution, being found in the liver and kidney, testis, and central nervous system [44,67,68,69,70]. The ratio of intracellular/extracellular lithium concentration is about five, suggesting a passive distribution of Li^+^ ion. The transport of lithium is in both directions across the cellular membrane through a Na^+^-dependent counter-transport system. Furthermore, Li^+^ ion enters the voltage-dependent Na^+^ channel, but the Na^+^-K^+^ pump mediates the lithium uptake but not its release in cells [71]. Therefore, neuronal cells maintained a ratio of internal-external Li^+^ ion below 1 mM in patients exposed to 1–2 mM of lithium [71].

### Lithium Transporters in Cancer

Regarding MCTs, evidence indicates that MCT1 and MCT4 are expressed in cancer cells, whereas scant information is available regarding MCT2 and MCT3 expression in cancer. Expression of MCT1 was reported for lung (non-small cell lung carcinomas, NSCLC), stomach, colon, bladder, prostate, breast, ovarian cervical, head and neck cancers, and gliomas [72,73,74,75,76,77,78]. On the other hand, MCT4 is expressed in colon, bladder, and breast cancers, and gliomas [74,76,77,78]. In addition, while MCT2 is expressed in NSCLS, colon, breast, and ovarian cancers [74,78], MCT3 is found in NSCLC [78] (Figure 1).

Furthermore, chaperone proteins of MCTs are important mediators in cancer. CD147 and CD44 function as chaperones of MCT1. In this way, the expression of MCT1 and CD147 correlated in bladder and ovarian carcinomas [78], whereas the expression of CD44 is associated with MCT1 expression in lung cancer [74]. Interestingly, the expression of MCT1 and CD147 in bladder tumors showed a poor prognosis [78]. However, when the expression of CD147 was downregulated by silencing using small interfering RNAs, an increase in the chemosensitivity to cisplatin was observed in bladder cell culture [78]. In prostate, breast, and lung cancers, the expression of MCT4 is associated with CD147 expression [74,78], suggesting a link between lithium transporters and their chaperone proteins in various cancers. In addition, evidence indicates that MCT’s differential expression depends on cancer progression. In this way, MCT1 is observed in healthy epithelium and malignant glands, while MCT2 was found in both benign to neoplastic lesions, but also in malignant glands. In contrast, the presence of MCT4 was observed in malignant glands with poor prognosis. These results suggest that MCT1 and MCT2 are involved in tumor maintenance and that MCT4 affects tumor aggressiveness [79]. Since MCTs have been proposed as biomarkers for cancer prognosis [79,80], their substrates such as LiAcAc, might be implicated in some cancer hallmarks, as discussed below. In addition, high expression levels of MCT1 and MCT4 are associated with increased invasion in lung cancer cells [81] and the inhibition of MCTs decreases the migration and invasion process in glioma [77], suggesting that MCTs have an important role in invasion and migration processes in cancer. Since lithium transporters have a pivotal role in some cancer hallmarks, the potential role of lithium in cancer biology needs to be addressed.

Cancer cells use several metabolic substrates such as glucose, glutamine, lipids, and lactate to grow and survive. Therefore, MCTs maintain a high-rate lactate transport, allowing glycolysis to preferentially fuel lactic fermentation [82]. On the other hand, acetoacetate is used to obtain energy, by oxidation through the TCA cycle generating acetyl-CoA [83], suggesting a possible function of lithium transporters in the metabolism of cancer cells; e.g., the BRAF^V600E^ mutation found in malignant melanomas, hairy cell leukemia, colorectal cancer, and multiple myeloma [84,85,86,87]. This mutation activates octamer transcription factor Oct-1, which promotes the expression of 3-hydroxy-3-methylglutaryl-CoA lyase (HMGCL) [88], resulting in an increase of acetoacetate levels, which promotes BRAF^V600E^ binding and activates the protein kinase MEK1 (mitogen-activated protein kinase/extracellular signal-regulated kinase) [88]. Furthermore, depletion of MEK1 reduced the colony formation of cellular culture, suggesting the important role of MEK1 in long-term proliferation and survival [89]. Therefore, HMGCL inhibitors or non-metabolizable acetoacetate compounds might be considered potential therapies for BRAF^V600E^ mutation cancers [88]. Interestingly, MCT1 is expressed in oxidative cancer cells and these types of transporters are capable of transporting lactate and LiAcAc [72,90]. In cells, the enzyme lactate dehydrogenase B (LDHB) oxidizes lactate into pyruvate, which inhibits the prolyl hydroxylases (PHDs) that catalyze the hydroxylation of the Hypoxia-Inducible Factor 1α (HIF-1α) on two proline residues, to initiate the proteasome-dependent degradation pathway of HIF-1α [90,91,92,93]. The question then arises, do lithium transports have a role in metabolism deregulation in cancer cells?

## 4. Lithium as a Specific Inhibitor of GSK3β

The chlorinated form of lithium inhibits Glycogen Synthase Kinase 3β (GSK3β), which is a serine/threonine protein kinase. GSK3β phosphorylates β-catenin in the Wnt metabolic pathway, leading to growth arrest [94,95]. Moreover, GSK3β participates in cellular proliferation and survival by activating nuclear factor ϰB-dependent gene transcription, suggesting an active role of GSK3β in tumorigenesis in pancreatic, colorectal, and prostate cancers, and gliomas [96,97,98,99,100]. Since GSK3β is involved in repressing Wnt/β-catenin signaling, but also in maintaining cell survival and proliferation via the NF-ϰB pathway, the inhibition of this dual kinase activity by lithium might have a positive effect on cancer hallmarks.

The expression of GSK3β is higher in colon cancer cells and in colorectal cancer patients in comparison to normal counterparts. Furthermore, the inhibition of GSK3β activity resulted in apoptosis induction and reduced proliferation in cell culture [99]. Interestingly, lithium is capable of inhibiting GSK3β; consequently, lithium is a potential therapeutic agent. Inhibition of GSK3β by LiCl takes place through two mechanisms. One of them is a competitive mechanism between lithium and magnesium for binding to GSK3β [101,102]. The other mechanism involves the role of LiCl in the phosphorylation of the serine-9, located in the N-terminal, which is the main regulator region of GSK3β [103]. However, GSK3β has contradictory functions in cancer cells. On one hand, GSK3β phosphorylates p53, leading to its inhibition [104]. In this way, GSK3β binds to p53 in camptothecin-treated neuroblastoma cells, leading to increased mitochondrial apoptosis signaling and the expression of p21 and Bax [105,106]. On the other hand, in lithium-treated cells, the nuclear complex GSK3β/p53 is disrupted, conferring chemoresistance to apoptosis in hepatoblastoma cells (HepG2) through GSK3β inhibition and repression of Cd95 expression [107].

Furthermore, when lithium blocks GSK3β activity, ovarian cancer cell proliferation was reduced and tumor suppressor activity was observed in nude mice inoculated with human ovarian cancer cells [108]. Moreover, proliferation was reduced in Li-treated human medullary thyroid TT cancer cells, due to the inactivation of GSK3β activity [109]. In addition, the inhibitory activity of lithium for GSK3β is linked to cell cycle arrest, by increasing cyclin-dependent kinase inhibitors such as p21, p27, and p15 [109]. In esophageal cancer, lithium decreases the proliferation of Eca-109 cells and affects the cell cycle, causing most cells to be in the G2/M phase [110].

The polycomb group gene *Bmi1* is increased in various cancers such as medulloblastoma and glioblastoma multiforme [111]. When *Bmi1* is downregulated, it triggers differentiation and reduces the expression of the stem cell markers *Sox2* and *Nestin* [111]. Since GSK3β activity is linked to *Bmi1*, inhibition of GSK3β activity by lithium triggered tumor cell differentiation, increased apoptosis, the disintegration of tumor spheroids, and decreased clonogenicity [111].

## 5. Effect of Lithium on Certain Cancer Hallmarks

### 5.1. Effect of Lithium on Apoptosis

The role of lithium as a proapoptotic or antiapoptotic compound is contradictory. In HepG2 cells, lithium confers resistance to the apoptosis induced by etoposide and camptothecin, by preventing the activation of caspase-8 and caspase-3 and by inhibiting the Bax translocation to mitochondria [107]. Therefore, lithium confers resistance to chemotherapy-induced apoptosis through GSK3β inhibition, disruption of GSK3β/p53 cooperation, and repression of CD95 expression [107]. Furthermore, in human malignant glioma cell lines (T98G and U87MG), non-proapoptotic effects regarding LiAcAc have been reported [112]. In hematological tumors (multiple myeloma), LiCl triggers cell cycle arrest and apoptosis by inhibition of GSK3β and the activation of the Wnt/β-catenin signaling pathway [24].

On the other hand, in human SH-SY5Y neuroblastoma cells, lithium inhibits GSK3β, contributing to antiapoptotic signaling mechanisms [113]. Furthermore, LiCl induces the decrease of proapoptotic proteins p53 and Bax, while increasing Bcl-2 (a major antiapoptotic protein that controls mitochondrial membrane permeability) levels [114]. In addition, the effects of mitochondrial toxins might be attenuated by the inhibition of GSK3β, which can be achieved by the activation of the phosphatidylinositol 3-kinase (PI3K)/Akt pathway, which is inhibited by GSK3β through Akt-mediated phosphorylation of Ser-9 of GSK3β [115] or by GSK3β direct inhibitors such as lithium, suggesting that the inhibition of GSK3β provides protection against the activation of the apoptosis-associated cysteine protease caspase-3 [116]. In human breast and colorectal cancer cell lines with oncogenic phosphatidylinositol 3-kinase subunit PIK3CA, lithium treatment reduced proliferation and reduced the GSK3β-target gene cyclin-D1, suggesting that GSK3β is an effector of oncogenic PIK3CA, suggesting lithium as a promising anti-neoplastic therapy against cancers harboring PIK3CA mutations [117].

In glioblastoma cells, low and higher LiCl doses had no effect on the expression of Bcl-2, procaspase-3, and PARP [118]. In contrast, in neuronal cells, Li blocks caspase-3 activation [119]. Therefore, cells of different origins (neuronal or glial) might have different results when treated with lithium. In cancer, several pathways might be dysregulated, and combinatory therapy with LiCl and pharmacological activators of the Notch1 pathway suppressed the hormonal secretion and reduces cell growth through apoptosis in medullary thyroid cancer (MTC) [120]. Lithium chloride enhances cell death induced by tumor necrosis factor-related apoptosis-inducing ligand (TRAIL) in human prostate cancer cells [121]. In addition to the TRAIL-mediated caspase-8 activation, LiCl enhances BID (BH3 interacting-domain death agonist) cleavage [121]. Li increases the activity of transcription factors such as activator protein-1 (AP-1) and cyclic-AMP response element binding protein (CREB) in vivo and in vitro [122] and activates the mitogen-activated protein (MAP) kinase pathway [123], which are involved in regulating apoptosis, cytokine production, and differentiation of promyelocytic leukemia cells [124]. Initiation and promotion of pancreatic ductal adenocarcinoma are linked to the Hedgehog-GLI signaling pathway, in which the GLI proteins are mediators and regulate cell differentiation [125]. Lithium decreases the expression and activity of glioma-associated oncogene-1 (GLI1) blocking cell proliferation and G1/S cell cycle progression, triggering apoptosis and reducing the tumorigenicity of pancreatic ductal adenocarcinoma cells [125]. In colorectal cancer cells, LiCl increased apoptosis and ROS levels, which could be mediated by the reduction of NF-κB expression [126]. Therefore, LiCl targets ROS/GSK3β/NF-κB pathways, resulting in increased apoptosis of colorectal cancer [126].

### 5.2. Effect of Lithium on Autophagy

Lithium induces autophagy and enhances the clearance of aggregated proteins, such as mutant huntingtin and α-synucleins neuronal precursor cell lines and non-neuronal cells. Interestingly, this effect is not mediated by GSK3β, but it was mediated by inhibition of IMPase, leading to free inositol depletion and reducing myo-inositol-1,4,5-triphosphate (IP_3_) levels, resulting in a novel form of autophagy induction that is independent of the mammalian target rapamycin (mTOR) [127,128]. In addition, the inhibition of IMPase and GSK3β showed opposite effects on autophagy. On one hand, Li induced the inhibition of IMPase at lower doses (Ki = 0.8 mM), enhancing autophagy [128]. On the other hand, the inhibition of GSK3β by Li^+^ ion downregulated autophagy (at higher doses, Ki ≅ 2mM) via activation of mTOR [129,130]. Furthermore, lithium chloride and other chemical forms induced autophagy by reducing the amount of pathological prion protein (PrP^Sc^) in prion-infected neuronal and non-neuronal cells [131]. Lithium carbonate-treated cells showed a high expression of LC3 and LAMP-1 (autophagy markers) and an increase in autophagic structure volume density, suggesting that the activation of autophagy by lithium may promote an increase of protein recycling in kidney proximal tubule cells [132].

Additionally, lithium carbonate promotes autophagic vacuole formation in hepatocellular carcinoma (HCC) cells in vivo, suggesting that lithium-mediated autophagy might be a novel approach for treating hepatocellular carcinoma [133]. Interestingly, in HCC, combinatory treatment of lithium carbonate and rapamycin increases autolysosome formation and the expression of autophagy markers such as LC3β and LAMP1 [134]. Further, 5 mM lithium carbonate produced an antitumor effect, by arresting the cell cycle in G2/M-phase and increasing the number of apoptotic HCC cells, as well as inducing autophagy [33]. In corneal endothelial cells, treatment with lithium carbonate upregulated the expression of *P62*, *Tmem74*, *Tm9sf1*, and *Tmem166*, indicating that lithium increased autophagy that may have contributed to increased endothelial cell survival in a mouse model of Fuchs endothelial corneal dystrophy [135]. Moreover, in human prostate cancer cells, lithium inhibited GSK3β, inducing cell death by modulating Bif-1, and in parallel induced an extensive autophagic response [136]. Blocking the autophagic response switched GSK3β-inhibition-induced necrosis to apoptotic cell death. Therefore, GSK3β promotes cell survival by modulating Bif-1-dependent autophagic response and cell death [136]. Finally, in acute promyelocytic leukemia, lithium chloride increases apoptosis and leads to cell cycle arrest at the G2/M phase. Although lithium increases the level of Ser9 phosphorylated GSK3β, it decreases the level of Akt1, inhibits c-Myc, and enhances cell death, with a concomitant increase in β-catenin. Therefore, LiCl promotes apoptosis through the Akt signaling pathway and cell cycle arrest by increasing phosphorylation of GSK3β [137]. In addition, lithium carbonate increases the number and volume of autophagic vacuoles and increases levels of LC3β, a biomarker of autophagy death [133]. Additionally, Li_2_CO_3_ induces apoptosis and arrests the cell cycle in the G2/M phase in hepatocellular carcinoma-29 cells [33].

### 5.3. Effect of Lithium on Tumor Growth, Tumor Proliferation, Tumor Invasion and Metastasis, and Cell Cycle Arrest

Tumor cell growth is controlled by various signaling pathways, such as PI3K/Akt and the raf-1/mitogen-regulated extracellular kinase (MEK)/extracellular regulated kinase (ERK) pathway (raf-1/MEK/ERK). LiCl decreases hormonal secretion and suppresses the growth of MTC in vitro and in vivo due to GSK3β inhibition [109]. The raf-1 activation induces GSK3β phosphorylation, therefore the inactivation of GSK3β by LiCl might be a mechanism of rat-1-induced growth arrest [109]. It has been reported that LiAcAc reduced cell growth and ATP concentration in seven human cancer cell lines, including colon and breast cancer cells [138]. Since uncoupling protein 2 (UCP2) was overexpressed in cancer cells, the authors suggested that the effect of LiAcAc might have been associated with an inefficient Randle cycle [138]. Furthermore, LiAcAc and LiCl inhibited bovine lymphocyte proliferation in vitro [139]. However, LiAcAc or LiCl did not affect the growth of glioma cell lines [112]. Moreover, LiAcAc had no effect on proliferation in the various breast cancer lines tested (BT20, BT474, HBL100, MCF-7, MDA-MB 231, MDA-MB 468, and T47D) [140].

Interestingly, low concentrations of LiCl (at 10, 20, 30, and 40 mM) promote the proliferation of mk3 and mk4 cells (metanephric mesenchyme cells), which play an essential role in nephron generation, by increasing the expression of the transcription factor *Six2* (*sine oculis* homeobox homolog 2 gene). In contrast, higher concentrations of LiCl (50 mM) inhibit proliferation and downregulate the expression of *Six2* [141]. Therefore, the potential anti-tumor effect of LiCl depends on its dose; low concentrations of LiCl induce proliferation via upregulating *Six2* expression, while high LiCl concentration decreases both proliferation and *Six2* expression, which is connected to the Wnt/β-catenin signaling pathway. Furthermore, in hippocampal neural stem/progenitor cells, low concentrations of LiCl increase cell proliferation [142]. In addition, LiCl induces G2 arrest in mouse embryonal cells and inhibits the activity of cyclin B/cdc2 kinase through interference with the dephosphorylation of Tyr-15 on cdc2 [143]. In addition, LiCl inhibits the cell proliferation of C6 glioma cells harboring an IDH2 (isocitrate dehydrogenase-2) mutation via GSK3β [144].

Lithium causes nuclear localization of the claudin-1 and β-catenin that form the apical junctional complex, which is associated with colorectal cancer progression. In addition, lithium reduces colony formation, inhibiting cell migration and cell proliferation through G2/M cell cycle arrest. In colorectal cancer, the aberrant activation of epidermal growth factor (EGF) disrupts the association between E-cadherin and β-catenin, increasing cell proliferation and migration [1]. EGFR activates downstream kinases that modulate the GSK3β phosphorylation and consequently activate the Wnt/β-catenin pathway [145,146]. Since LiCl inhibited the tumorigenic effects of EGF, this cation might have tumor suppressor activity in colorectal cancer [147]. LiCl inhibits multiple-myeloma proliferation in a dose-dependent manner and triggers cell cycle arrest and apoptosis through inhibition of GSK3β, which is a crucial mediator of the Wnt/β-catenin pathway [24]. GSK3β regulates cadherin-11 at transcriptional and translational levels. Lithium inhibition of GSK3β suppresses cadherin-11, which is involved in cell–cell adhesion, cancer cell invasion, and metastasis of cancer [148]. Moreover, dedifferentiation, invasion, and metastasis are associated with E-cadherin loss [149]. Although the loss of E-cadherin or β-catenin, along with the downregulation of WNT7, has a poor prognosis in lung cancers, lithium led to the induction of E-cadherin and WNT7 in an inositol-independent manner [149].

In colon cancer, lithium reduces the expression of transforming growth factor-β-induced protein (TGFBIp) by inhibiting Smad3 phosphorylation through GSK3β inhibition [150]. In addition, Li inhibits cell migration of lymphatic endothelial cells and prevents metastasis by inhibiting TGFBIp-induced tumor lymphangiogenesis, resulting in the inhibition of colon cancer metastasis [150]. The effects of lithium on migration are greater than its effects on viability and occur at lower doses and earlier time points. Moreover, the migration effects of Li are reversible, suggesting that migration blockade is not due to cytotoxicity [151]. Moreover, GSK3β and GSK3α inhibition by lithium reduces migration and invasion of glioma cells [151]. In primary bovine aortic endothelial cells, LiCl induced cell cycle arrest in the G2/M phase via over-expression of cyclin-dependent kinase inhibitor (p21^Cip^) in a p53-dependent pathway, suggesting that lithium stabilizes β-catenin and activates p53 [152]. In the presence of LiCl, stabilized β-catenin is translocated into the nucleus, where it associates with transcription factors such as TCF (T-cell factor), leading to the expression of genes involved in cell adhesion and cell proliferation [153,154,155,156].

Genome-wide screening has been used to determine that LiCl inhibits cell proliferation, reduces replication, and arrests the cell cycle in multiple prostate cancer cells [157]. In prostate cancer, LiCl participates in the reduction of DNA replication and S-phase cell cycle arrest, by disrupting the interaction of E2F factor with DNA, decreasing the expression of the cell division cycle 6 (*cdc6*), *cyclin A*, *cyclin E*, and *cdc25C*, which are regulated by E2F [157]. Furthermore, the effects of LiCl might be associated with GSK3β inhibition [157]. In addition, promyelocytic leukemia cells evade death by increasing the levels of IL-2 and IL-10, at the same time as decreasing the amount of TNF-α and IL-6 [158]. In this context, LiCl induces cell death by promoting the expression of TNF-α and IL-6, while inhibiting IL-2 and IL-10 production [158].

The anti-cancer properties of lithium salts are summarized in Figure 2.

## 6. Anti-Inflammatory Activity of Lithium

In some types of cancer, before the cell becomes malignant, inflammatory conditions occur. Therefore, inflammation is associated with the proliferation and survival of cancer cells. In addition, this promotes angiogenesis and metastasis [159]. Since inflammation is associated with the development and progression of cancer, inflammatory biomarkers such as pro-inflammatory cytokines, tumor necrosis factor (TNF), interleukins 1 and 6, and chemokines can be used to monitor cancer progression [160]. In this way, the anti-inflammatory effect of lithium might be an alternative for prevention and monitoring of cancer. Furthermore, in animal models, LiCl treatment induced therapeutic effects on inflammation-mediated skeletal muscle wasting such as sepsis-induced muscle atrophy and cancer cachexia [161].

The anti-inflammatory activity of lithium results from its strong inhibitory effect on GSK3β, leading to a reduction in TNF-α production via attenuated activation of NF-κB and JNK signaling cascades [124] and induction of cytokine IL-10 [162]. The anti-inflammatory activity of lithium is due to the reduction of cyclooxygenase-2 expression, induction of the nitric oxide synthase expression, inhibition of interleukin 1β and TNFα, and increased production of IL-2 and IL-10 [163,164]. Furthermore, LiCl increases the levels of TNF-induced IL-6 in murine skin [165]. During tumor growth, toxic products are generated that cause damage to organs distal to the tumor site. In this way, lithium carbonate had a protective effect on podocytes during tumor growth induced by inoculation of hepatocellular carcinoma-29 cells in CBA mice, suggesting that lithium may have a protective role in organs during tumor growth [166].

## 7. Synergism of Lithium with Standard Cancer Therapies

High concentrations of lithium (4–5 mM) inhibit inositol 1-monophosphatase, resulting in a reduction of phosphatidylinositol turnover. However, these high concentrations of Li^+^ ion have toxic effects in patients [14]. Reported serum Li^+^ ion levels for therapy were 0.8 to 1.2 mEq/L (0.8–1.2 mM) and the cytotoxicity effect began at 1.5 to 2 mEq/L (1.5–2 mM) [167]. Interestingly, 1 mM of LiCl had no effect on cellular metabolism or cell cycle but its combination with cisplatin or paclitaxel reduced metabolic activity in a serous ovarian cancer cell line (OVCA 433) or primary cultures. However, LiCl by itself or in combination with cisplatin or paclitaxel had no significant effect on cellular proliferation, suggesting that the effects of LiCl are mainly related to cellular metabolism rather than proliferation [167]. These results observed in vitro contradict a phase II study of 15 patients with low-grade neuroendocrine tumors treated with LiCl that showed non-positive results in terms of response evaluation criteria in solid tumors and in overall survival, as well as GSK3β phosphorylation [168], suggesting that therapeutic usage of lithium depends on various factors, such as tumor nature and aggressiveness, lithium salts, and doses that allowed GSK3β phosphorylation, among others.

Conventional cancer therapies such as chemotherapy and radiotherapy inhibit cell division, causing neurocognitive deterioration. Preliminary studies have shown that lithium has a neuroprotective role in cancer patients undergoing radiotherapy or chemotherapy treatment, improving their quality of life [169]. Combination of LiCl with a low dose or IC_50_ dose of etoposide enhances apoptosis and reduces the percentage of cells in the G2/M phase in prostate cancer LNCap cells [19]. In pancreatic ductal adenocarcinoma cells, lithium increases the anti-tumoral efficacy of gemcitabine [125]. Additionally, histone deacetylase inhibitors with lithium resulted in an effective combinatory therapy, due to inducing the Notch1 pathway and inhibiting GSK3β activity, respectively. [120]. Furthermore, temozolomide combined with a low dose of LiCl (1.2 mM) induced cell death in TP53wt glioma cells by inhibiting GSK3, promoting the nuclear translocation of the nuclear factor of activated T cells (NFAT1), and upregulating the expression of Fas/FasL, resulting in a potential combinatorial therapy for glioblastoma treatment [23]. In addition, breast cancer cells (T47D) treated with LiCl (20 mM) had increased cellular radiosensitivity through decreasing Mre11 mRNA levels, resulting in decreased DNA repair in T47D monolayer and spheroid cells [170]. Neurite growth-promoting factor 2 (NEGF2), also known as Midkine (MK), is a cytokine that promotes cell growth, differentiation, survival, gene expression, and drug resistance. MK, PI3K, and GSK3β inhibitors together with LiCl could be a very effective treatment modality for tumors with a high expression of these molecules [171]. Pre-treatment of breast cancer cells with LiCl induced a synergistic cytotoxicity effect with mitomycin C (MMC), resulting in apoptosis via HMGB1 and *Bax* signaling, but LiCl also prevented MMC-induced necrosis [25].

Radiotherapy has various consequences in children, such as debilitating cognitive decline, partly linked to impaired neurogenesis. In this context, LiCl has a protective role in the juvenile brain against radiotherapy and improves the quality of life of these children [142]. In addition, LiCl is a promising drug for rescuing neurogenesis in adult and juvenile brains after irradiation, suggesting that Li^+^ prevents the debilitating cognitive effects of cranial radiotherapy [172,173]. Since lithium is a specific radio-sensitizer for tumor cells, it can be used in combination with radiotherapy [174]. LiCl has an immunomodulatory capability for regulating titanium nanoparticle-stimulated inflammatory responses through the inhibition of the MAPK signaling pathway [175]. In colorectal cancer, the effects of LiCl on the upregulation of pro-apoptotic proteins and the downregulation of expression of a protein associated with survival were enhanced by high-energy photon irradiation [31]. Therefore, lithium might be used as a sensitizer for colon cancer cells, to improve the results of radiation therapy. On the other hand, pre-treatment with lithium in thyroid cancer patients before radioiodine therapy demonstrated no improvement in radioiodine uptake in thyroid tissue [176].

Chemotherapy-induced peripheral neuropathy (CIPN) is an adverse effect of cancer treatment using taxanes, such as paclitaxel. Adding lithium to taxane treatment prevents calcium overloading in mitochondria, resulting in the prevention of calpain activation and preventing the loss of cell functions related to the progression of peripheral neuropathy [177], suggesting that lithium could be used to prevent the adverse effects of cancer treatment with only taxane administration [178]. In addition, transient receptor potential V4 (TRPV4) is involved in the initiation of CIPN. Evidence suggests that prolongated exposure to paclitaxel increases the acute effects of TRPV4 expression, currents, and calcium fluxes, while pre-treatment with lithium decreases the expression of TRPV4, currents, and calcium fluxes, suggesting that lithium prevents the CIPN caused by paclitaxel [179].

In a systems biology analysis, the interactomes of lithium-sensitive enzymes and the pathways associated with several types of cancer suggested that Li affects the incidence and progression of cancer [180]. Despite evidence recommending designing and conducting clinical trials of lithium for cancer, efforts prioritize the use of Li for bipolar disorder. Patients with bipolar disorders who were treated with Li showed a significantly lower cancer risk, suggesting an association between Li and cancer risk in a dose-dependent manner [181]. Furthermore, lithium (300 mg) administrated once daily for 5 days in each course of chemotherapy to 36 breast cancer patients was not effective in preventing the adverse effects of this cancer treatment [182]. Another clinical trial conducted with nine patients with acute myeloid leukemia (AML), who were orally treated with lithium carbonate (300 mg), resulted in the inhibition of GSK3, which induces differentiation of leukemic cells and could target AML stem cells [183].

## 8. Recent Clinical Studies

Based on the information available (http://www.clinicaltrials.gov (accessed on 20 January 2023)), thirteen clinical trials have been conducted (Table 2). Some of these clinical studies include intervention with lithium and anti-cancer drugs, such as oxaliplatin, capecitabine, temozolomide, bevacizumab, and aripiprazole. Recent clinical studies have focused on the potential use of lithium in colorectal neoplasm, stomach neoplasm, esophageal neoplasm, medullary thyroid cancer, neuroendocrine tumors, prostate cancer, small cell lung cancer, breast cancer, adult acute lymphoblastic leukemia in remission, adult acute myeloid leukemia in remission, osteosarcoma, brain and central nervous system tumors, and familial adenomatous polyposis. Additionally, lithium carbonate was employed in combination with radiation protocols. The results of these clinical trials could provide us with important information for understanding the potential synergism of lithium with current therapies.

## 9. Trends in the Administration of Lithium

Medicine has adopted nanotechnology strategies for drug administration. Similarly, nanosized lithium being administered as nanoparticles or other nanostructures has been reported. Injection of lithium carbonate nanoparticles in tumors causes cell necrosis, destruction of the vascular bed, and attraction of neutrophils and macrophages to the tumor [189]. During the development of hepatocarcinoma-29 in mice, injection of nanosized lithium carbonate was essential for the functioning of macrophages [20]. Interestingly, Li_2_CO_3_ nanoparticles protected vital organs such as the heart and lungs of CBA mice with hepatocellular carcinoma from the damage caused by secondary products of lipid peroxidation and increased macrophages and neutrophils within the tumor, decreased blood micro-vessel density, and increased the tumor necrosis rate [190].

Moreover, nanoparticles of lithium citrate and carbonate suppressed the proliferation of hepatoma-29 cells in lower concentrations than their officinal forms [191]. Previous reports indicated that hepatocarcinoma-29 cells are more sensitive to nanoparticles of lithium salts such as citrate and carbonate in comparison to their original forms [192]. Furthermore, nanoparticles of lithium citrate mainly induce apoptosis, whereas lithium carbonate nanostructures induce apoptosis and autophagic death of hepatocellular carcinoma cells [22]. In addition, nanosized lithium citrate induces apoptosis, whereas nanoparticles of lithium carbonate induce apoptosis and autophagic cell death of HCC cells [22]. Liposomal lithium carbonate nanostructures have a longer half-life, equal efficacy, and better therapeutic dose maintenance in blood compared to their standard-sized counterparts for bipolar disorder treatment [193].

In addition, leaf extract of *Zanthoxylum armatum*, a tropical medicinal shrub, was used as a reducing agent to synthesize titanium dioxide (TiO_2_) nanoparticles. The TiO_2_ nanomaterial exerted cytotoxicity and induced apoptosis through lipid peroxidation in breast cancer cells. The anti-tumor efficacy and safety of TiO_2_ nanomaterial were evaluated in vivo using a subcutaneous 4T1 breast BALB/c mouse tumor model, displaying significantly less cardiotoxicity and body weight loss compared to doxorubicin [194]. These findings open new possibilities for the preparation of nanosized lithium using plant extracts as reducing agents. Furthermore, a lithium carbonate in situ gelling system appears to be a promising vehicle for Li^+^ ion administration for bipolar disorders, but its pharmacodynamics and pharmacokinetic properties need to be addressed [195]. However, these nano-drugs also need to be investigated for cancer treatment. To avoid fluctuations in the plasma concentration of lithium, a push-pull osmotic pump was developed for lithium carbonate delivery, to maintain Li^+^ ion at therapeutic levels [196]. Cellular death results from apoptotic and necrotic processes that have an anticancer effect [197], but apoptosis rather than necrosis is the accepted pathway of cellular death under the immune system control [198]. A mesoporous silica matrix containing Li showed the potential for induction of apoptosis, while reducing necrosis [199].

Lithium formate monohydrate was irradiated by X-rays, electrons, and protons for use as a dosimeter in radiotherapy [200]. In addition, pre-treatment with LiCl induced an increased cellular reactive oxidative species (ROS) level, resulting in a better therapeutic effect of adipose-derived stem cells in intervertebral disc degeneration-related disease [201].

The biocompatibility and multifunctionality of Li-doped bioceramics with polymers provided new biomaterials used in 3D printing for clinical applications of tissue engineering [202]. Furthermore, lithium-ferrite bioactive glasses prepared using the sol-gel methodology were used for hyperthermia therapy. Lithium-ferrite bioactive glasses provide temperatures up to 47.2 °C, making this material a candidate for cancer treatment using hyperthermia [203]. However, no in vitro assays have been published

Furthermore, lithium has been used for the synthesis of new nanomaterials with potential applications in cancer therapy. Layered metal oxides such as MoO_3_ and WO_3_ treated with lithium transformed white-colored micrometer-long MoO_3_ nanobelts into blue-colored short, thin, defective, interlayer gap-expanded MoO_3−x_ nanobelts with strong NIR-II absorption. With this surface modification, the MoO_3−x_ nanobelts could be a promising nanomaterial for photoacoustic imaging-guided photothermal therapy, resulting in an efficient cancer cell ablation and tumor eradication under irradiation from a 1064 nm laser [204]. Lithium intercalation-assisted exfoliation is an approach for the preparation of nanomaterials with applications in noninvasive photothermal therapy [205]. To overcome the tumor hypoxia-induced by oxygen-dependent photodynamic therapy, ultrathin vermiculite nanosheets were synthesized by lithium-ion intercalation for aggregation-induced emission photosensitizer loading, to produce a nanodevice inspired by bonsai [206]. In addition, LiCl alters the sodium and potassium levels, altering the membrane potential in human breast cancer cells (MCF-7), and suggesting that LiCl exerts a dose-dependent biphasic effect on MCF-7 cells by altering the apoptotic/antiapoptotic balance [207].

## 10. Contradictory Effects of Lithium

Prolonged treatment with lithium causes renal, endocrine, and dermatological side effects. Four weeks of therapy with lithium caused renal tubular concentration defects, polyuria, polydipsia, and nephrogenic diabetes insipidus [208]. Administration of Li_2_CO_3_ induced a cutaneous adverse reaction [209] and parathyroid dysfunction [210]. Furthermore, lithium treatment caused a higher prevalence of hypercalcemia in a cohort of bipolar patients [211]. In addition, there is a possibility of teratogenicity activity with lithium treatment [212].

The inhibition of GSK3β using lithium has contradictory effects. On one hand, chemoresistance to apoptosis is observed, but on the other hand, it induces tumor suppressor activity and reduces proliferation. Those effects depend on the type of cancer, the origin of cells, and doses. The anti-cancer effect of lithium has been reported for colorectal cancer, neuroblastoma cells, ovarian cancer cells, medullary thyroid TT cancer cells, esophageal cancer, medulloblastoma, and glioblastoma multiforme, hematological tumors (multiple myeloma), breast cancer cells, MTC, prostate cancer cells, pancreatic ductal adenocarcinoma cells, corneal endothelial cells, and colon cancer cells. However, opposite effects of lithium have been reported for neuroblastoma, hepatoblastoma cells, and malignant glioma cell lines.

Lithium induces apoptosis in promyelocytic leukemia cells, but a small fraction of these cells remain viable and survive the cytotoxic effect of this cation, resulting in altered gene expression patterns [213], suggesting cell tolerance to lithium in culture [214]. Resistance to lithium is a consequence of its activity. Lithium appears to induce apoptosis by increasing the *bax*/*bcl-2* ratio (*bax*: pro-apoptotic gene and *bcl-2*: anti-apoptotic gene). Lithium resistance then results when this ratio is reversed, leading to lithium resistance accompanied by DNA fragmentation, and decreasing cell viability [158].

## 11. Conclusions

The expression of transporters of lithium is linked to various types of cancer, such as NSCLC, colon, stomach, bladder, prostate, breast, ovarian, cervical, head and neck cancer, and gliomas. In both in vivo and in vitro models of cancer, lithium is capable of activating the Wnt/β-catenin pathway. The GSK3β kinase is activated when a secreted glycoprotein (Wnt) binds to its receptor. GSK3β is inhibited by lithium through Ser-9 phosphorylation, stabilizing free β-catenin in the cytoplasm. Therefore, the GSK3β might have anti- or pro-apoptotic activity depending upon the cell type. According to genomics and proteomics data, lithium salts such as citrate, carbonate, chloride, and acetoacetate may stimulate antitumor effects, such as apoptosis, autophagy, cell death, tumor growth, tumor proliferation, invasion, and metastasis and arrest the cell cycle in in vivo and in vitro models (Figure 2). Since Li^+^ ion has potential for cancer treatment, the selection of the delivery form of lithium, such as nanosized or original forms, plays a key role in triggering the antitumor response, which depends on its transport into the cell, the cancer type, doses, and administration. Li^+^ ion selectivity for tumor cells, however, needs to be further investigated, as well as doses and administration.

## Figures and Tables

**Figure 1 life-13-00537-f001:**
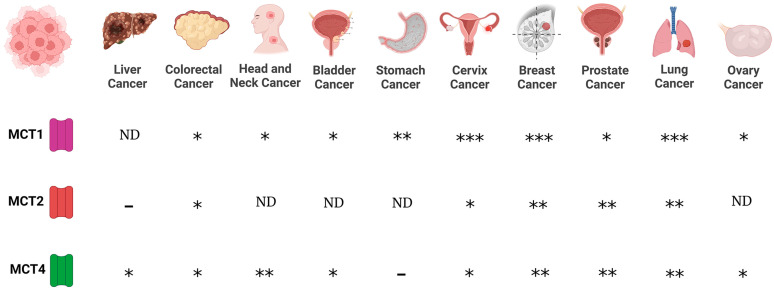
Expression of monocarboxylate transporters (MCTs) in various cancers. MCTs are involved in the passive transport of lithium acetoacetate across cell membranes, and their abundance is associated with cancer. (-), no expression; (*), medium expression; (**), high expression; (***), very high expression level; ND, not data.

**Figure 2 life-13-00537-f002:**
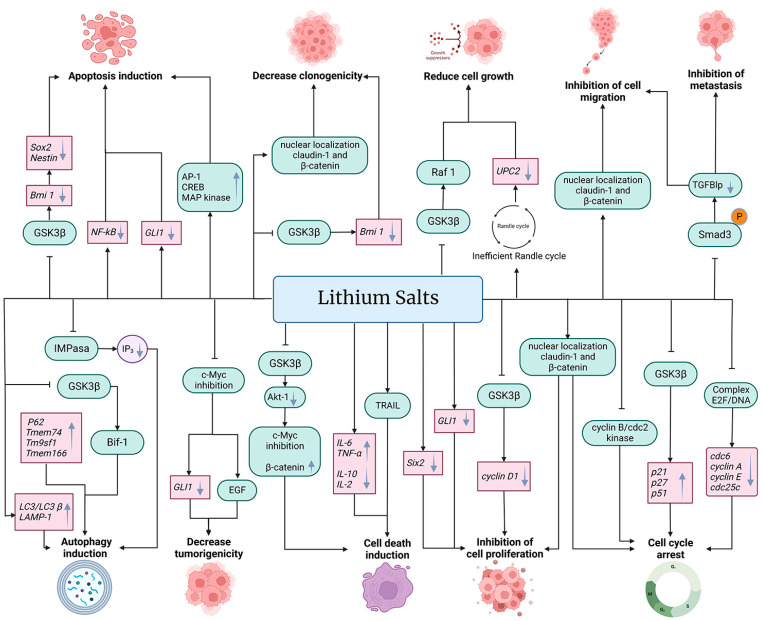
Effects of lithium on cancer cells. Treatment with salts of lithium such as lithium acetoacetate (LiAcAc), lithium chloride (LiCl), lithium citrate (Li_3_C_6_H_5_O_7_), and lithium carbonate (Li_2_CO_3_) inhibit cell proliferation, cell migration, metastasis, and cell growth, in addition to decreasing tumorigenicity and clonogenicity, arresting the cell cycle and inducing autophagy, cell death, and apoptosis. Lithium affects cancer cells at the transcriptional (gene expression level) (pink boxes) and at post-transcriptional (protein function, expression, and cell localization) (green boxes) levels. In addition, lithium affects metabolic pathways and other metabolic products (purple circle). Black arrows (→) indicate the effect of lithium on the molecule (gray downward arrow (↓) denotes decrease or down-regulation, and gray upward arrow (↑) denotes increase or up-regulation). The truncated black line (⊥) indicates an inhibitory effect of lithium.

**Table 1 life-13-00537-t001:** Recent preclinical studies of lithium in cancer treatment.

Type of Salt	Type of Cancer	Experimental Model	Dose of Lithium	Outcomes of the Study	Ref.
LiCl	Lymph Node Carcinoma of the Prostate cells	Human prostate carcinoma LNCaP cells	2.5, 10, 25, and 45 mM	LiCl showed a cytotoxic effect in a dose- and time-dependent manner (*p* < 0.001). LiCl increased apoptosis of Lymph Node Carcinoma of the Prostate cells in the presence of etoposide, which is S- and G2-phase-specific drug.	[19]
Li_2_CO_3_ particles	Hepatocarcinoma 29	Male CBA mice	Li_2_CO_3_ particles 10–20 nm in diameter (0.058 mg)	Nanosized lithium carbonate particles were inessential for the function of peritoneal macrophages.	[20]
LiCl	Gastrointestinal tract cancers (colorectal carcinoma)	Female balb/c mice	Lithium chloride (200 mg/kg)	The efficacy of chemotherapeutic agents was improved with lithium, by enhancing non-apoptotic cell death.	[21]
Li_3_C_6_H_5_O_7_ and Li_2_CO_3_ particles	Hepatocellular carcinoma	Cell line HCC-29 (hepatocarcinoma)	5 mM lithium salts	Nanosized lithium citrate mainly induced apoptosis, whereas lithium carbonate induced apoptosis and autophagic cell death.	[22]
LiCl	Glioblastoma	Human TP53wt U87 and TP53mut U251glioblastoma cells with methylated MGMT promoters	1.2 mM	Low doses of lithium in combination with temozolomide induced glioma cell death via NFAT1/FasL signaling.	[23]
LiCl	Multiple myeloma	Human multiple myeloma cell lines (RPMI-8226 and U266)	40 mM	Lithium inhibited cell proliferation and induced cell cycle arrest.	[24]
LiCl	Breast cancer	MDA-MB-231 breast cancer cells	ND	Induced autophagy in breast cancer cells treated with LiCl and mitomycin C.	[25]
LiAcAc LiCl	Neuroblastoma, renal cell carcinoma	Human embryonic kidney cell line HEK293, human normal dermal fibroblasts (HDFn) and RCC 786-O cells	2.5, 5, and 10 mM LiAcAc and with equimolar concentrations of lithium chloride (LiCl)	LiAcAc and LiCl affected the growth of all cell lines, either negatively or positively.	[18]
LiCl	Breast cancer	SUM159 MCF7 MDA-MB-231 and HEK293T	20 mM	In triple-negative breast cancers, GSK3β was upregulated and correlated with worse survival in patients. Inhibition of GSK3β via LiCl decreased the expression of markers of the mesenchymal phenotype, indicating its ability to impede the process of epithelial-mesenchymal transition (EMT).	[26]
LiCl	Breast cancer	MCF7 MDA-MB-231	30 and 20 mM	Treatment with LiCl increased the active GSK3β protein, and DNA damage, and decreased survival, independently of estrogen receptor status in breast cancer cells exposed to ionizing radiation.	[27]
LiCl	Lung cancer	A549	50 ng/ml	The protein expression levels of PD-L1 were upregulated after treatment with LiCl. PD-L1 is the major ligand of PD-1 and is expressed in a variety of tumors, including in NSCLC. Overexpression of PD-L1 was implicated in tumor immunity, and inhibition of PD-L1 enhanced antitumor immunity by preventing tumor cells from escaping host immune responses.	[28]
LiCl	Nasopharyngeal carcinoma	C666-1,5-8F and SUNE-1	20 mM	LiCl was used to activate Wnt/β-catenin, which is correlated with the EMT process in cancer.	[29]
LiAcAc	Breast cancer	Breast cancer cell lines (MCF7, MDA-MB-231, Hs578T)	3 mM, 10 mM, or 30 mM	Cytotoxic effects of LiAcAc treatment were significantly similar to those caused by LiCl.	[30]
LiCl	Colorectal cancer	T88 primary colon cancer cells from a patient with sporadic colon cancer.	30 mM	LiCl could be used to sensitize colon cancer cells to radiation therapy.	[31]
LiCl	Human head and neck squamous cell carcinoma (HNSCC)	HNSCC SCC9, SCC25, A253, DETROIT562 & Fadu cell lines	11.45 mM, 15.24 mM, 14.56 mM, 11.4 mM, 5.53 mM	LiCl reduced the proliferation and colony formation capabilities of HNSCC cell lines and influenced wound closure. LiCl increased the inhibitory Ser9 phosphorylation of GSK3β, leading to an increment of the GLI3 repressor form; inhibiting HH-GLI pathway activity.	[32]
Li_2_CO_3_	Hepatocellular carcinoma	HCC-29 cell line	5 mM	Li_2_CO_3_ arrested the cell cycle in the G2/M-phase and induced apoptosis and autophagy in HCC-29 cells. The upregulation of autophagy markers LC3B and LAMP1 was assessed using the presence of Li_2_CO_3_.	[33]
LiCl	Hepatocellular carcinoma	HepG2 cell line	20 mM	Antitumor effects of KIF18B-siRNA, which targets KIF18B, an important in pairing and separation of chromosomes, could be reversed by LiCl treatment, through the Wnt/β-catenin pathway	[34]
LiCl	Myeloma	Human MM cell line RPMI8226	20 mmol/L	MiR-135b is involved in the development and progression of various cancers. In this study, LiCl was used to activate the Wnt/β-catenin pathway, reversing the effects of downregulating miR-135b on the proliferation, migration, invasion, and apoptosis of MM cells.	[35]
LiCl	Ameloblastoma, an odontogenic epithelial tumor	AM-1 cell line and human ameloblastoma cells	1 and 4 mM	Lithium chloride (LiCl) increased the size and decreased the proliferation of cells, and expression of *Sox2*.	[36]
Li_2_CO_3_	Skin melanoma	B16 melanoma cells	ND	Li_2_CO_3_ inhibited cell proliferation and stimulated cell death in melanoma cells through induction of autophagy and apoptosis.	[37]

ND: No data available.

**Table 2 life-13-00537-t002:** Clinical trials elucidating the value of lithium in cancer clinic ^1^.

NCT Number	Title	Status	Conditions	Interventions	Characteristics	Population	Results	Ref.
NCT03153280	Dose Escalation Study of Lithium with Oxaliplatin and Capecitabine in Advanced Oesophago-Gastric or Colorectal Cancer (Lithium)	Recruiting	Colorectal Neoplasms Stomach Neoplasm Esophageal Neoplasms	Drug: Lithium Drug: Oxaliplatin Drug: Capecitabine	Study type: Interventional Phase: Phase 1	Enrollment: 24 Age: 18 Years and older (Adult, Older Adult) Sex: All	No data were analyzed, reported, or published on this study.	ND
NCT00582712	An Initial Study of Lithium in Patients with Medullary Thyroid Cancer	Terminated	Medullary Thyroid Cancer	Drug: Lithium carbonate	Study Type: Interventional Phase: Phase 2	Enrollment: 5 Age: 18 Years and older (Adult, Older Adult) Sex: All	No data were analyzed, reported, or published on this study.	ND
NCT00501540	Lithium for Low-Grade Neuroendocrine Tumors	Completed	Neuroendocrine Tumors	Drug: Lithium Carbonate	Study Type: Interventional Phase: Phase 2	Enrollment: 15 Age: 18 Years and older (Adult, Older Adult) Sex: All	Based on the pre- and post-treatment tumor biopsies, lithium did not potently inhibit GSK-3β at the serum levels used to treat bipolar disorders.	[168]
NCT02198859	Evaluation of lithium and its effect on clinically localized prostate cancer	Completed	Prostate Cancer	Drug: Lithium Carbonate	Study Type: Interventional Phase: Phase 1	Enrollment: 11 Age: 18 Years and older (Adult, Older Adult) Sex: Male	No study results posted on ClinicalTrials.gov for this study	ND
NCT00251316	Effect of Lithium Carbonate on Low-Dose Radioiodine Therapy in Early Thyroid Cancer	Completed	Thyroid Cancer Differentiated Thyroid Carcinoma	Drug: Lithium Carbonate Drug: Placebo	Study Type: Interventional Phase: Phase 2	Enrollment: 34 Age: 16 Years and older (Child, Adult, Older Adult) Sex: All	Lithium carbonate improved the success rate of postsurgical ablation of thyroid carcinoma.	[184]
NCT01486459	A Feasibility Trial Using Lithium as A Neuroprotective Agent in Patients Undergoing Prophylactic Cranial Irradiation for Small Cell Lung Cancer (TULIP)	Terminated	Small Cell Lung Cancer	Drug: Lithium	Study Type: Interventional Phase: Not Applicable	Enrollment: 7 Age: 18 Years and older (Adult, Older Adult) Sex: All	No study results posted on ClinicalTrials.gov (accessed on 20 January 2023) for this study	ND
NCT05221593	Efficacy of Lithium Against Chemotherapy Induced Neutropenia in Breast Cancer Patients	Completed	Neutropenia	Drug: Lithium Carbonate	Study Type: Interventional Phase: Phase 3	Enrollment: 50 Age: Child, Adult, Older Adult Sex: All	No study results posted on ClinicalTrials.gov for this study	ND
NCT00469937	Ph I Study of Lithium During Whole Brain Radiotherapy for Patients with Brain Metastases	Terminated	Brain and Central Nervous System Tumors Cognitive/Functional Effects Neurotoxicity Solid Tumor	Drug: lithium carbonate Procedure: cognitive assessment Procedure: quality of-life assessment Radiation: radiation therapy	Study Type: Interventional Phase: Phase 1	Enrollment: 9 Age: 18 Years and older (Adult, Older Adult) Sex: All	No study results posted on ClinicalTrials.gov for this study	ND
NCT01553916	Neuroprotective Effects of Lithium in Patients with Small Cell Lung Cancer Undergoing Radiation Therapy to the Brain	Completed	Small Cell Lung Carcinoma	Drug: Lithium Carbonate Radiation: Prophylactic cranial irradiation	Study Type: Interventional Phase: Phase 1 Phase 2	Enrollment: 19 Age: 18 Years and older (Adult, Older Adult) Sex: All	No study results posted on ClinicalTrials.gov for this study	ND
NCT01105702	Temodar (Temozolomide), Bevacizumab, Lithium and Radiation for High Grade Glioma	Terminated	Brain Cancer	Drug: Temozolomide Drug: Bevacizumab Drug: Lithium Carbonate Radiation: Radiation	Study Type: Interventional Phase: Phase 2	Enrollment: 28 Age: 18 Years and older (Adult, Older Adult) Sex: All	Antiangiogenic agents produced high radiographic response rates.	[185]
NCT00408681	Lithium Carbonate in Treating Patients with Acute Intestinal Graft-Versus-Host-Disease (GVHD) After Donor Stem Cell Transplant	Completed	Accelerated Phase Chronic Myelogenous Leukemia Adult Acute Lymphoblastic Leukemia in Remission Adult Acute Myeloid Leukemia in Remission Adult Acute Myeloid Leukemia with 11q23 (MLL) Abnormalities Adult Acute Myeloid Leukemia with Inv(16) (p13; q22) Adult Acute Myeloid Leukemia with t(15;17) (q22;q12) Adult Acute Myeloid Leukemia with t(16;16) (p13;q22) Adult Acute Myeloid Leukemia with t(8;21) (q22;q22) Atypical Chronic Myeloid Leukemia, Breakpoint Cluster Regional Translocation (BCR-ABL) Negative Blastic Phase Chronic Myelogenous Leukemia	Drug: lithium carbonate Other: laboratory biomarker analysis	Study Type: Interventional Phase: Not Applicable	Enrollment: 20 Age: 18 Years to 75 Years (Adult, Older Adult) Sex: All	8 of 12 patients (67%) had a complete remission (CR) of GVHD and survived more than 1 year (median 5 years) when lithium administration was started promptly within 3 days of endoscopic diagnosis of denuded mucosa. When lithium was started promptly and less than 7 days from salvage therapy for refractory GVHD, 8 of 10 patients (80%) had a CR and survived more than 1 year.	[186]
NCT05402891	The CHAMP-study: The Chemo preventive Effect of Lithium in Familial Adenomatous Polyposis	Enrolling by invitation	Familial Adenomatous Polyposis	Drug: Lithium Carbonate	Study Type: Interventional Phase: Phase 2	Enrollment: 12 Age: 18 Years to 35 Years (Adult) Sex: Al	Lithium had a chemo-preventive effect in familial Adenomatous Polyposis.	[187]
NCT01669369	Clinical Trial of Lithium Carbonate Combined with Neoadjuvant Chemotherapy to Treat Osteosarcoma	Recruiting	Osteosarcoma	Drug: Lithium Carbonate Drug: Placebo	Study Type: Interventional Phase: Phase 4	Enrollment: 400 Age: 8 Years to 70 Years (Child, Adult, Older Adult) Sex: All	Lithium Carbonate combined with neo-adjuvant chemotherapy improved the prognosis of osteosarcoma	[188]
NCT01820624	Lithium Carbonate and Tretinoin in Treating Patients with Relapsed or Refractory Acute Myeloid Leukemia	Completed	Adult Acute Megakaryoblastic Leukemia (M7) Adult Acute Monoblastic Leukemia (M5a) Adult Acute Monocytic Leukemia (M5b)-	Drug: tretinoin Drug: lithium carbonate Other: laboratory biomarker analysis	Study type: Interventional Phase: Phase 1	Enrollment: 12 Age: 18 years and older (Adult, Older Adult) Sex: All	GSK3 inhibition using lithium carbonate resulted in acute myeloid leukemia cell differentiation and may be a novel therapeutic strategy in this disease	[183]

^1^ Data available at https://clinicaltrials.gov/ct2/results?cond=Cancer&term=lithium&cntry=&state=&city=&dist= (accessed on 20 January 2023). ND: No data.

## Data Availability

Not applicable.

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
