# Peer review of "Lithium: A Promising Anticancer Agent"

_life, 2023, doi:10.3390/life13020537_

Round 1
Reviewer 1 Report
This is an interesting study and the authors have collected a useful dataset. However, in my opinion, there are some points that need to be improved:
· Introduction section: this part is weakly written; I suggest including a simple introduction about cancer as a disease, highlighting the main cancer hallmarks, explaining about Warburg effect, and then giving details about lithium and its potential as an anticancer agent.
· The authors need to add to a table that summarizes the preclinical studies. The column of the table could be included the type of cancer, experimental model, activity dose of lithium, outcomes of the study, and so on. It would be more suitable to use sources published in the past 10 years.
· Since the authors are discussing how lithium is a potential anticancer agent, a section of the recent clinical studies should be included in this article.
· Line 300 to 305: are mk3 and mk4 normal cell lines? If they are what is the point of mentioning them? Line 304-305 doesn’t make sense, according to the authors a low concentration of LiCl proliferation was enhanced by upregulation of the Six2 gene.
· In lines 350-351: the authors describe the effect of LiCl on TNF-α and interleukins, and contradict this information in figure 2. As well, lines 347-351 are supposed to be in the next section.
· In the section on anti-inflammatory effect: it would be more convenient if the authors have demonstrated the correlation between cancer and inflammation and how lithium could reduce inflammation biomarkers.
· It might be more suitable if add a section on the contradictory effect of lithium instead of being brought into the text once in a while.
Author Response
We thank the reviewers for their significant effort in helping us with their comments and suggestions. We have carefully analyzed and considered all their comments and observations to clarify their doubts. We are confident that our manuscript has now been improved.
In the uploaded file we address point by point the comments raised by the reviewers and indicate the changes made in the manuscript.

Reviewer 2 Report
1. The manuscript is very general the authors are suggested to rewrite with precisely the role of Li in bipolar disorders and cancer therapy.
2. Add a paragraph about the Li transport to cells in cancer therapy.
Author Response
We thank the reviewers for their significant effort in helping us with their comments and suggestions. We have carefully analyzed and considered all their comments and observations to clarify their doubts. We are confident that our manuscript has now been improved.
In the uploaded file, we address point by point the comments raised by the reviewers and indicate the changes made in the manuscript.

Reviewer 3 Report
This review provides a general overview of lithium, from its transport into the cells to its innovative administration forms based on genomic, transcriptomic, and proteomic data. It reported a revision of the scientific evidence of the potential use of lithium as a therapeutic drug to treat several types of cancer. In my opinion the writing of this article needs major revision as following comments:
1. It is recommended to check the manuscript carefully before submission to avoid typos and grammar problems.
2. It is necessary to add more literature citations in the past 3-5 years. Moreover, some papers are just accumulated but there are not enough descriptions of the research contents, the author's views, and illustrations about the theoretical innovation. Especially the existing problems and development trends of lithium in cancer treatment are strongly recommended to supplement.
3. The writing and logic of the title and secondary title need to be improved. There are some problems in the logical expression of the article thus misleading the readers. It is suggested to adjust the overall framework of the article according to the classification of acts upon pathways.
4. "Synergism of lithium with standard cancer therapies", and "Trends in the administration of lithium", these two sections are too simple expressed. More current researches and evidences are required to support a complete review.
Author Response

(The authors gave the same response as above.)

Round 2
Reviewer 1 Report
The authors have done a significant effort to improve the manuscript; however, there are some points that need to be considered:
Regarding the title of the manuscript, it would be more convenient as “Lithium: a promising anticancer agent” (optional).
The headline of the section “lithium” is supposed to be changed to something like “The effect of lithium on cancer progression: preclinical studies”.
In table 1:
· Reference (26) the outcomes of the study need to be written briefly
· Reference (30) the study is not related to the content of the table (the experimental model is not a cancer model)
· References (32), (34), (38), and (41) the authors need to revise the section on “type of cancer” the text is irrelevant to the section.
· Reference (39) the outcome of the study doesn’t include the effect of lithium.
In table 2:
· It would be better to replace the title of the study with a references section.
· The results of the study need to be inserted in the table.
Author Response
The point-by-point responses to the reviewer's comments are in the uploaded file.

Reviewer 2 Report
The authors have extensively revised the manuscript but still need some minor changes which is stated below.
1. I think the title is not appropriate which should be like, Lithium: a promising cancer therapeutic agent or tool or regimen.
2. L105, the ion Li+ shares the transport and permeation pathways with other ions. The word ion should be after Li.
3. L167, Cancer cells use several metabolic substrates such as glucose, glutamine, lipids, and lactate. please complete the sentence.
4. L432, please replaced the word actual with suitable word like conventional.
5. Please add recent literature using metal for cancer therapy like
https://pubmed.ncbi.nlm.nih.gov/34201266/
www.sciencedirect.com
Author Response

(The authors gave the same response as above.)

Reviewer 3 Report
This review looks much better and comprehensive after careful revision. I agree to accept the manuscript in the current form.
Author Response

(The authors gave the same response as above.)
